# Dementia Literacy and Willingness to Dementia Screening

**DOI:** 10.3390/ijerph17218134

**Published:** 2020-11-04

**Authors:** Yoko Aihara, Kiyoshi Maeda

**Affiliations:** Kobe Gakuin University, Kobe City 651-2180, Hyogo, Japan; maedak@reha.kobegakuin.ac.jp

**Keywords:** dementia literacy, health literacy, information, screening, willingness

## Abstract

The increasing prevalence of delayed and missed diagnoses for dementia constitutes major public concern. In this regard, inadequate knowledge and poor understanding of the condition may create a barrier to timely dementia screening. This cross-sectional study assessed dementia literacy, then identified the association between dementia literacy and willingness to undergo routine dementia screening among community-dwelling older adults in two urban areas of Japan. More specifically, structured questionnaires were distributed to a total of 854 individuals aged ≥ 65 years. A multivariate logistic regression was then used to explore the factors associated with dementia literacy and willingness to undergo routine dementia screening. Results showed that younger respondents and respondents who received dementia information from television/radio and/or paper-based sources were more likely to have high dementia literacy. While less than half of participants were willing to undergo routine dementia screening, those with higher dementia literacy were more willing to do so (albeit, not a statistically significant difference). Although there are pros and cons to routine dementia screening, it is necessary to implement such a system to detect dementia and cognitive impairment. Further, assessments should also attempt to gain information about individual beliefs and understandings related to dementia information.

## 1. Introduction

A recent systematic review of the global burden of diseases reported that 43.1 million people are living with dementia worldwide, with the number more than doubling from 1990 to 2016 [1]. Although the magnitude of public issues related to dementia is increasing, other studies have revealed that approximately 62% of dementia cases are undetected [2]. Similarly, anywhere from one-half to two-thirds of early-stage dementia cases are likely missed during diagnosis [3]. In order to improve the lives of people with dementia and their caregivers while decreasing the community-wide impacts of the disease, the World Health Organization (WHO) announced the “global action plan on the public health response to dementia 2017–2025.” One of the plan’s most important strategies involves raising public awareness about dementia, thus fostering a more accurate understanding [4]. In this regard, adequate knowledge may reduce the stigmatization of people with dementia while also resulting in better early recognition practices. 

Health literacy is defined as the range of cognitive and social skills that enable people to obtain, understand, and use information for the purpose of enhancing their health and well-being and engage in healthcare-related decision-making [5]. Health literacy also refers to one’s knowledge, motivations, and competencies when accessing, understanding, appraising, and applying health information [6]. Individuals with lower health literacy are unable to adequately understand and access health information, which may lead to negative attitudes about cancer screening [7]. Several studies have also reported that older adults with lower health literacy tend to have poorer health outcomes, lower preventive service compliance, and higher healthcare utilization [8,9,10]. In this context, improved dementia literacy may increase early diagnosis of dementia and result in more timely support for people with dementia.

As mentioned above, there are both pros and cons to routine dementia screening. For example, the US Preventive Service Task Force concluded that there was a lacking of direct evidence for the benefits of early screening to detect cognitive impairments in older adults [11]. Dementia screening may also not be effective for healthy individuals due to issues stemming from both misdiagnosis and overdiagnosis, which can have significant long-term effects including stigmatization, the loss of employment and/or autonomy, economic problems, and additional burdens placed on health systems that do not have the capacity to respond to increased demands resulting from screening [12]. On the other hand, a national early dementia detection program may improve quality-adjusted life expectancy while also reducing the costs of screening for older persons [13]. Although interventions intended to cure dementia have been implemented, recent studies have also indicated that approximately one-third of dementia cases are likely preventable [14]. As the majority of people living with dementia remains undiagnosed, it is highly important to increase public awareness about the methods for identifying the diseases, thereby resulting in appropriate treatments and timely social support.

Japan has one of the highest rates of population aging in the world. In this context, current estimates indicate that more than 7 million people aged 65 years and older will have dementia by 2025, with the total costs of the disease expected to increase 1.6 times between 2014 and 2060 [15]. For those reasons, early dementia diagnosis has become part of a national dementia strategy. Several local governments have also initiated subsidized systems that offer free dementia screenings at hospitals and clinics. A previous study investigated the intention to use the subsidized system among community-dwelling older adults and found that while only 20% of participants knew about the system, 60% were willing to use free dementia screenings [16]. Following this evidence, we hypothesized that additional information and adequate knowledge about dementia would be associated with willingness to undergo dementia screening. Thus, this study aimed to assess dementia literacy, then identified the association between dementia literacy and willingness to undergo regular dementia screening among community-dwelling older adults in urban areas of Japan.

## 2. Materials and Methods

### 2.1. Study Location and Study Participants

This study was conducted in an area adjacent to the cities of Akashi and Kobe in Japan. Both initiated subsidized systems for dementia screening for older adults in 2019 fiscal year. In the same year, an initial survey was conducted among all community-dwelling persons aged 65 years and above who lived in this area (details are given elsewhere) [16]. Among the 2269 total respondents, 1165 agreed to participate in a follow-up survey. Self-administered questionnaires were thus distributed to each of these individuals, resulting in 854 response (rate of 73%). The sample size was calculated using a formula for the interval estimation of the population proportion, in which the outcome variable was set as dementia literacy. We followed information from a previous study reporting that 55% of older adults living in urban China had adequate dementia literacy [17]. Using an expected sampling error of 0.05, a confidence interval of 0.95, and a potential proportion of adequate dementia literacy = 55%, we deemed that a minimum sample of 381 was sufficient. The survey was conducted from November 2019 to March 2020.

### 2.2. Variables

The outcomes of the variables were set as dementia literacy and willingness to undergo regular dementia screening. The dementia literacy component was developed based on eight questions developed by Zhang et al. [17] specifically regarding the symptoms, prevalence, nature, and treatment methods for dementia. Prevalence and treatment options were modified to fit the Japanese context. Internal consistency reliability of the literacy questions was not high but acceptable (Cronbach’s alpha coefficient = 0.57). Willingness to undergo dementia screening was determined by asking participants whether they were willing to undergo regular (annual) dementia screening (answers were selected from options of “yes,” “no,” and “do not know”). Participants who indicated they were unwilling to engage in these screenings were also asked to select from the following reasons: “shameful to be diagnosed with dementia,” “fear of being diagnosed with dementia,” “there is no cure for the disease,” “bothersome to visit the clinic,” “economic burdens,” “annoyance to family members,” “do not know which doctor can be consulted,” and others.

As relevant information sources are related to dementia literacy [18], participants were asked which sources they use to learn about the disease. This included options of television/radio, verbal information (family members and friends), paper-based information (books, magazine/newspaper, and local paper), professional information (medical staff and classes), and Internet. We then assessed intellectual activity levels by asking participants how often they chat with family members and/or friends (almost every day, 2–3 times per week, 1–2 times per month, or rarely) and how often they read newspapers/magazines/books (almost every day, sometimes, or rarely). Cognitive impairments and depressive symptoms are also associated with dementia literacy and willingness to undergo regular dementia screenings [17,19]. We, therefore, assessed cognitive impairments using the Dementia Assessment Sheet for Community-based Integrated Care System 21 items (DASC-21), which has been previously deemed sufficiently reliable and valid for detecting dementia [20]. Depressive symptoms were then assessed based on five items from the Geriatric Depression Scale (GDS-5) [21], the Japanese version of which has been previously deemed sufficiently valid [22]. Self-rated health status was determined based on 4-point Likert scale consisting of “very good,” “good,” “fairly poor,” or “poor.” We also asked respondents for basic demographic information, including items on age, gender, family structure, educational level, and whether they had primary care physicians.

### 2.3. Analysis

Descriptive statistics were applied to calculate the proportions and numbers of each variable. Each question on dementia literacy was scored as either correct (1 point) or incorrect/do not know (0 points) (total scores ranging from 0 to 8 points). The average dementia literacy score was 4.2, thus categorized as “high dementia literacy” based on a threshold of ≥4 points, with “low dementia literacy” being determined at scores <4 points. Willingness to undergo dementia screening was scored as either “yes” (1 point) or “no/do not know” (0 points). A chi-squared test was then conducted to identify the factors associated with dementia literacy and willingness to undergo regular dementia screenings while a multiple logistic regression model was applied to control for confounding factors. All statistical analyses were performed using the STATA 16.0 software (StataCorp., College Station, Texas, USA) and statistical significance was set to 5%.

### 2.4. Ethics

All participants received written explanations of the study protocol. We then obtained written, informed consent from those who agreed to participate. This study received ethical approval by the Ethics Committee of Kobe Gakuin University (protocol number Sourin-18–16, approved on 6 February 2019).

## 3. Results

### 3.1. Factors Associated with Dementia Literacy

The average participant age was 78.2 years (range of 65–100), with 53% being female. Among all participants, 14% were assessed as possibly having dementia (DASC-21 ≥ 31), while 28% had depressive symptoms (GDS-5 ≥ 2).

Only 1.7% of the respondents answered all dementia literacy questions correctly. The question with the highest rate of correct answers was that on memory loss from dementia (88.2%), while the lowest rate was for the item on the prevalence of dementia in Japan (22.7%) (Table 1). The majority of respondents obtained dementia-related information from television (75%) and newspapers/magazines (52%). On the other hand, 9% of the respondents answered that they did not receive any dementia-related information.

Table 2 shows the factors associated with dementia literacy. High literacy was associated with younger ages, higher educational levels, living with family members, using more dementia-related information sources, suspectable depression, and higher frequencies of chatting and reading.

A multivariate logistic regression showed that younger respondents and those who obtained dementia information from television/radio and/or paper-based sources were more likely to have high levels of dementia literacy (Table 3).

### 3.2. Willingness to Undergo Regular Dementia Screening and Dementia Literacy

Among all respondents, 41% were willing to undergo regular dementia screening, with more than half being unwilling or undecided. The main reasons for unwillingness were “bothersome to visit the clinic” (42%) and “do not know which doctors can be consulted” (32%) (Table 4).

A univariate logistic regression model showed that respondents with higher dementia literacy were more likely to accept regular dementia screenings than those with low dementia literacy (odds ratio (OR) = 1.40, 95% confidence interval (CI) = 1.02, 1.92). However, there were no statistically significant differences after controlling for other variables (OR = 1.36, 95% CI = 0.94, 1.93). Older adults with poor self-rated health were more likely to undergo regular dementia screening than those with better self-rated health (OR = 0.64, 95%CI = 0.42, 0.97) (Table 5).

## 4. Discussion

This study assessed dementia literacy and willingness to undergo regular dementia screenings among community-dwelling older adults. Results showed that respondents of younger ages had higher dementia literacy after controlling for educational level, depression, and cognitive impairment. A systematic review reports that older age was strongly associated with limited health literacy [23], while another study suggested that age-related changes in cognitive function that are not captured by dementia screening may affect health literacy levels among the older adults [24]. Decline in hearing and vision functions with aging may also be related to limited dementia literacy. As dementia is strongly associated with advanced age, it is especially important to improve dementia literacy among older populations. In this regard, educational interventions may help with dementia management as older persons with limited health literacy tend to perceive more barriers to communication [25]. Active learning education has been proven effective for improving health literacy and health behaviors [26] and use of simple language and visual elements (e.g., pictures and photos) has been supported. 

In this study, we found that the majority of participants gathered dementia-related information from a variety of sources, and diversity of sources was significantly associated with higher dementia literacy. Among the variety of sources, respondents who obtained dementia information from television/radio and/or paper-based sources had higher overall levels of dementia literacy than those who did not. Similar findings were reported by a previous study targeting older adults in Vietnam, which showed that television and Internet usage were associated with higher health literacy [27]. An empirical study showed that the use of paper-based nutritional information was associated with higher nutrition literacy among older people [28]. Television and radio are easy media to access information for older people. The global action plan for dementia recommends collaboration between the media and relevant stakeholders during awareness campaigns. In this context, dementia-related information is broadcasted frequently to raise public awareness about dementia in Japan. A qualitative study found that participants with adequate functional health literacy identified a variety of information sources [29], while exposure to dementia-related information from multiple sources were associated with greater dementia knowledge [30]. In sum, dementia literacy can substantially be increased through use of multiple information sources.

Our initial survey found that 60% of community-dwelling older adults intended to undergo dementia screenings using the subsidized system for dementia diagnosis [16]. Although it may be assumed that free dementia screenings are attractive for older people, this follow-up survey revealed that less than half of respondents wished to undergo regular dementia screenings. Similar to this study’s findings, 49% of older adults living in retirement communities in the U.S. stated that they would agree to routine screening for memory problems [31]. Many participants said they did not wish to undergo routine dementia screenings due to the burdens associated with clinic visitation and/or not knowing which physicians can be consulted. It is also known that early-stage dementia and mild cognitive impairments are not risk factor for life. Given these conditions, many people may not believe it is important to diagnose early dementia. A previous study conducted in Australia reported that many participants were optimistic about the prognosis for dementia and that more than half of the respondents did not worry about getting dementia, while 85% of them answered they would not recognize the early symptoms of Alzheimer disease [32]. This study also showed that low correct response rates were found for items on the nature (28.4%) and prevalence of dementia (22.7%). This may be because many older adults are interested in knowing how to prevent dementia, but there is relatively little interest in early recognition of symptoms. However, a systematic review reported that attitudes toward undergoing dementia screening were diverse and fragmented [33]. A qualitative study conducted in the UK found that acceptance of dementia screening depended on a variety of reasons [34]. To promote screening acceptance, further research needs to explore the reasons for acceptance or refusal to undergo routine dementia screening.

This study showed that higher dementia literacy was marginally associated with an increased preference for regular dementia screening. A previous literature review reported that low levels of awareness about screening led to misunderstandings about both the reason for dementia tests and the implication of test results [33]. Another study similarly found that individuals with low health literacy were less likely to retain cancer screening information [35], with other research indicating that low health literacy may constitute a barrier to regular screening participation [36]. Although this study did not find a statistically significant association between dementia literacy and the willingness to undergo dementia screening, raising public awareness and provision of screening information are part of the key strategies for promoting accessibility of dementia screening.

Further, participants with poor self-rated health status were more willing to undergo regular dementia screenings than those with better self-rated health status. A study involving older adults in the USA also found that unhealthy individuals were more willing to undergo routine dementia screenings [31], while another qualitative study reported that persons’ existing health state and perceived susceptibility to illness may impact the acceptability of dementia screenings [37]. Along with this study’s results, it thus appears that older adults with poor health status may have increased concerns about the need for long-term care than healthy individuals, which likely influences their decision to undergo routine dementia screenings. Although there is currently insufficient evidence about the effectiveness of routine dementia screening [11], multiple benefits and comparatively few/minor harms are associated with specific screening tests for dementia [38]. Increasing participation of healthy individuals in dementia screening and provision of information regarding benefits of timely diagnosis of dementia are needed.

This study also had limitations. First, respondents agreed to participate in the follow-up survey after receiving information on the study purpose. In this regard, these may have generally been more interested in dementia and were, thus, more active in collecting dementia information. Sampling bias may have, therefore, led to an overestimation of dementia literacy. Second, content validity of the instrument for measuring dementia literacy was not high (Cronbach’s alpha= 0.57). Various instruments and aspects of dementia literacy have been used for dementia research [39]; thus, further research needs to accurately measure dementia literacy to better understand association between literacy and dementia. Missing data were excluded in multiple regression analyses because the numbers of incomplete data were less than 10% and the results of the analyses may not be biased [40]. However, if the amounts of missing values are large, we need to handle missing values for reducing bias in terms of estimation of parameters of interests [40]. Moreover, we only assessed willingness to undergo regular screenings and did not assess actual behavior in this regard. Further study is needed to identify the association between dementia literacy and participation in dementia screening.

## 5. Conclusions

More than half of this study’s participants were unwilling to undergo routine dementia screenings, with many referring to the burdens associated with clinic visitation and the lack of sufficient information about screening. It is, therefore, important to increase access to adequate dementia information, which should raise general awareness about dementia and, thus, improve timely diagnosis of dementia. Further, sources such as television/radio and/or paper-based information may be effective for improving dementia literacy, which influences decisions to undergo dementia screening and receive advance care. Although there are pros and cons to routine dementia screening, a system of detecting dementia and cognitive impairment should be implemented to better understand individual beliefs and understandings related to the disease. Further research also needs to clarify which interventions are effective in increasing dementia literacy as well as changing perspectives on undergoing dementia screening among community-dwelling older adults.

## Figures and Tables

**Table 1 ijerph-17-08134-t001:** Questions of dementia literacy.

Questions	Answers	Correct Rate
Q1. In the following opinion, which one do you NOT agree?	1-Dementia is a disease **2-Dementia is not a disease, it is a normal aging process** 3-Dementia is common among older adults 4-Dementia does not mean disease, but rather refers to a variety of disease 5-I do not know	28.4%
Q2. Currently, among people aged older 65 years, one in how many persons are living with dementia?	1-One in two persons 2-One in four persons 3-One in five persons **4-One in seven persons** 5-I do not know	22.7%
Q3. In the following options, which one is likely to be a symptom of dementia?	1-Headache 2-Faint 3-Dizziness **4-Bad memory** 5-I do not know	88.2%
Q4. Which is the most common difficulty would people with dementia meet?	1-Learning difficulty 2-Difficulty in work 3-Difficulty in self-care **4-All of above (all 1–3 options)** 5-I do not know	43.5%
Q5. Which is the most common symptom of dementia?	**1-It is easier to forget the recent events than the past events** 2-It is easier to forget the past events than the recent events 3-It is easy to forget both past events and recent events 4-None of above 5-I do not know	61.5%
Q6. Which of the following is NOT a symptom ofDementia?	1-Easy to get lost 2-It cannot remember someone’s name 3-Often forget to return the things that they borrowed back **4-Do not forget their own things** 5-I do not know	58.8%
Q7. If you or your family members get dementia, which doctor you would not seek for help?	1-Neurosurgeon **2-Neurologist** 3-Internist (General physician) **4-Psychologist** 5-I do not know	46.3%
Q8. In the following opinions related to dementia, which one do you agree?	1-Dementia is not a disease, treatment is not needed 2-Although dementia is a disease, treatment is not necessary 3-Dementia cannot be cured, treatment is not needed **4-Dementia is a disease, treatment is needed** 5-I do now know	71.8%

Note: Answers shown in bold are correct answer.

**Table 2 ijerph-17-08134-t002:** A chi-square test to identify factors associated with dementia literacy (*n* = 775).

Variables	High Literacy (*n* = 519)	Low Literacy (*n* = 256)	*p*-Value
Age			<0.001
65–74 years	189 (80.1%)	47 (19.9%)	
75+ years	330 (61.2%)	209 (38.8%)	
Gender			0.52
Male	252 (68.1%)	118 (31.9%)	
Female	267 (65.9%)	138 (34.1%)	
Educational level			0.005
Elementary school/junior high school	78 (56.1%)	61 (43.9%)	
High school	230 (69.1%)	103 (30.9%)	
University/post-graduate school	210 (70.2%)	89 (29.8%)	
Missing	1 (25.0%)	3 (75.0%)	
Family structure			0.03
Living alone	153 (60.7%)	99 (39.3%)	
Living with family members	278 (69.9%)	156 (30.1%)	
Missing	4 (80.0%)	1 (20.0%)	
Average number of information sources	2.79	2.12	<0.001
Cognitive impairments			0.11
DASC-21 < 31	426 (68.7%)	194 (31.3%)	
DASC-21 ≥ 31	58 (58.6%)	41 (41.4%)	
Missing	35 (62.5%)	21 (37.5%)	
Depressive symptoms			0.004
GDS-5 < 2	372 (71.3%)	150 (28.7%)	
GDS-5 ≥ 2	137 (60.6%)	89 (39.4%)	
Missing	10 (37.0%)	17 (63.0%)	
Frequency of chat with family/friends			0.003
Almost everyday	366 (71.1%)	149 (28.9%)	
2–3 times per week	103 (60.2%)	68 (39.8%)	
1–2 times per month or less	49 (55.7%)	39 (44.3%)	
Missing	1 (100%)	0	
Frequency of reading			0.006
Almost everyday	406 (69.8%)	176 (30.2%)	
Sometimes	73 (62.4%)	44 (37.6%)	
Rarely	40 (52.6%)	36 (47.4%)	

**Table 3 ijerph-17-08134-t003:** Multivariate logistic regression model to identify factors associated with dementia literacy (*n* = 775).

Variables	Odds Ratio	95% Confidence Interval
Age (year)	0.94	0.91, 0.96
Gender		
Male	Ref.	
Female	0.87	0.60, 1.25
Educational level		
Elementary school/Junior high school	Ref.	
High school	1.07	0.66, 1.75
University/Post-graduate school	1.03	0.62, 1.72
Family structure		
Living alone	Ref.	
Living with family members	1.41	0.96, 2.08
Dementia information sources		
Television/radio	1.54	1.02, 2.34
Verbal information	1.15	0.81, 1.65
Internet	1.98	0.99, 4.00
Profession	1.58	0.93, 2.69
Paper-based	1.56	1.07, 2.29
Cognitive impairments		
DASC-21 < 31	Ref.	
DASC-21 ≥ 31	0.98	0.58, 1.64
Depressive symptoms		
GDS-5 < 2	Ref.	
GDS-5 ≥ 2	0.79	0.53, 1.18
Frequency of chat with family/friends	1.12	0.86, 1.46
Frequency of reading	1.25	0.81, 1.96

Willingness to undergo regular dementia screening and dementia literacy.

**Table 4 ijerph-17-08134-t004:** The reasons of unwilling to undergo regular dementia screening.

Reasons	%
Shameful to be diagnosed with dementia	4.5
Fear of being diagnosed with dementia	17
There is no cure for the disease	9.5
Bothersome to visit clinic	41.8
Economic burdens	9.9
Annoyance to family members	4.5
Do not know which doctor can be consulted	31.5
Others	26.2

**Table 5 ijerph-17-08134-t005:** Univariate and multivariate logistic regression for factors associated with undergoing regular dementia screenings (*n* = 807).

Variables	Univariate Logistic	Multivariate Logistic
	OR (95% CI)	OR (95% CI)
Age (year)	1.01 (0.99, 1.03)	1.01 (0.98, 1.03)
Gender		
Male	Ref.	Ref.
Female	1.12 (0.85, 1.49)	1.19 (0.86, 1.65)
Educational level		
Elementary school/Junior high school	Ref.	Ref.
High school	1.56 (1.03, 2.34)	1.46 (0.90, 2.36)
College/University	1.47 (0.96, 2.19)	1.49 (0.91, 2.43)
Family structure		
Living alone	Ref.	Ref.
Living with family members	0.78 (0.58, 1.05)	0.72 (0.51, 1.03)
Dementia literacy		
Low	Ref.	Ref.
High	1.40 (1.02, 1.92)	1.35 (0.94, 1.93)
Cognitive impairment		
DASC-21 < 31	Ref.	Ref.
DASC-21 ≥ 31	1.10 (0.72, 1.67)	1.12 (0.68, 1.84)
Depressive symptoms		
GDS-5 < 2	Ref.	Ref.
GDS-5 ≥ 2	0.94 (0.68, 1.29)	0.81 (0.54, 1.20)
Having primary care physicians		
No	Ref.	Ref.
Yes	1.32 (0.81, 2.14)	1.37 (0.78, 2.39)
Self-rated health		
Fairy poor/poor	Ref.	Ref.
Very good/good	0.76 (0.54, 1.06)	0.64 (0.42, 0.98)

OR = odds ratio, 95% CI = 95% confidence interval.

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
