# Peer review of "Dementia Literacy and Willingness to Dementia Screening"

_ijerph, 2020, doi:10.3390/ijerph17218134_

Round 1

Reviewer 1 Report

It was pleasure to review this manuscript.

The article left a good impression. It addresses a issue that is well-founded in the introduction. Introduction short, but focus on main ideas related with topic, i.e., starting from some trends of dementia, next defining health literacy, dementia literacy, lacking evidence of early dementia screening, and finally explaining research context. My comments. On the end of introduction it was hypothesized that additional information and adequate knowledge about dementia would increase the number of people willing to undergo dementia screening (page 2, lines 62-64). This is too strong hypothesis, it seems that you would like to measure causal relationship. But study was not experimental. You just measure association among dementia literacy and willingness to undergo regular dementia screening.

Also I missed research aim on the end of introduction. But it was stated in abstract.

Material and methods

Research participants and sampling procedures were explained quite clear.

Instruments was also explained quite clear, but I missed some information. Specifically, lacking information regarding adaptation of literacy instrument.It was just mentioned that this instrument was modified to fit Japan context. Also no information about reliability of this instrument.

The Geriatric Depression Scale (GDS-5) - was it validated in Japan?

Discussion clear. Maybe too much focus on information from television/ radio and paper-based sources.

Conclusions seems more  like some recommendation, but I liked such style than just repetition of main results. 

Author Response

Thank you for your valuable comments. Please see the attachment.

Reviewer 2 Report

As a qualitative reviewer I am not able to comment on the methodology or design. However, as a qualitative researcher reading the study raises questions as to what we can learn from this survey study.

First of all I wonder why such a small number of respondents answered ALL the questions on dementia literacy, was this that they did not understand all the questions? 

There is nothing surprising about the fact that high literacy was associated with higher education, and other factors - except perhaps that they are younger - why does being younger link to higher literacy and is this about engagement - and if so it is relevant to the study in terms of considering how to engage older adults with understandings in relation to dementia - a key finding and recommendation from the study.

The authors may wish to draw further on the other most notable finding from the study that participants were unwilling to undergo regular dementia screening because it was bothersome to attend clinic. It seems to me that this needs either unpacking further or as a recommendation for further research. Almost more powerful than furthering publicity and information campaigns. Why is it bothersome? 

How could screening be made more accessible - this seems the most pertinent of all the findings and yet is not made as much of by the authors. 

Also it appears that there is evidence from other geographical contexts that shows that those with worse health are more likely to attend screenings. Therefore how can those in good health be persuaded? What benefit would this be to them? And is this something for further study in order to develop campaigns that reach this group of participants if this practice is seen as being important in the desire to manage the growth in those living with dementia.

For me, as a qualitative researcher it raises questions about the need to understand why people choose to seek help in the first place, or issues around attitudes to self-care and health promotion activities, and individual responsibilities to these, particularly if they are in good health and feel it is an unnecessary burden. Therefore the paper suggests the authors need to further consider the limits of what we can learn from this survey and what direction it might point further research towards. 

On a further note, the paper requires more proofreading, and I noted an error with a word missing on line 62. 

Author Response

Thank you for your valuable comments. 
